# Wealth and inequality gradients for the detection and control of hypertension in older individuals in middle-income economies around 2007-2015

**María Fernanda García[1], Philipp Hessel[2,3], Paul Rodríguez-Lesmes[1]** *

**1** School of Economics, Universidad del Rosario, Bogotá, Colombia, **2** Escuela de Gobierno, Universidad de los Andes, Bogotá, Colombia, **3** Swiss Tropical and Public Health Institute, Basel, Switzerland

* paul.rodriguez@urosario.edu.co

**Data Availability Statement:** Minimum replication scripts and data is avaialable at: https://github.com/androdri1/HBP_wealthgradient Full data is available at the SAGE project webpage: https://apps.who.int/

## Abstract

Socioeconomic inequalities in the detection and treatment of non-communicable diseases represent a challenge for healthcare systems in middle-income countries (MICs) in the context of population ageing. This challenge is particularly pressing regarding hypertension due to its increasing prevalence among older individuals in MICs, especially among those with lower socioeconomic status (SES). Using comparative data for China, Colombia, Ghana, India, Mexico, Russia and South Africa, we systematically assess the association between SES, measured in the form of a wealth index, and hypertension detection and control around the years 2007-15. Furthermore, we determine what observable factors, such as socio-demographic and health characteristics, explain existing SES-related inequalities in hypertension detection and control using a Blinder-Oaxaca decomposition. Results show that the prevalence of undetected hypertension is significantly associated with lower SES. For uncontrolled hypertension, there is evidence of a significant gradient in three of the six countries at the time the data were collected. Differences between rural and urban areas as well as lower and higher educated individuals account for the largest proportion of SES-inequalities in hypertension detection and control at the time. Improved access to primary healthcare in MICs since then may have contributed to a reduction in health inequalities in detection and treatment of hypertension. However, whether this indeed has been the case remains to be investigated.

## 1 Introduction

The burden of non-communicable diseases (NCDs) in middle-income countries (MICs) is growing due to demographic and lifestyle changes. Already 85% of all premature NCDs deaths are in MICs [1]. This is a pressing concern for low-resourced economies which have yet to deal with transmissible diseases as well as the lack of adequate human capital and physical resources of their healthcare systems [2], therefore threatening to achieve the Sustainable

healthinfo/systems/surveydata/index.php/catalog/sage/about And upon request to Ministerio de Salud y Protección Social de Colombia: Dirección de Epidemiología y Demografía, which should be done through the webpage: https://tramites.minsalud.gov.co/tramitesservicios/.

**Funding:** GADC project by the CIHR/IDRC [grant number 108442-001] Fulbright-Colciencias and Colombia Cientifica – Alianza EFI 60185 contract FP44842- 220-2018, funded by The World Bank through the Scientific Ecosystems, managed by the Colombian Ministry of Science, Technology and Innovation (MINCIENCIAS). The funders had no role in study design, data collection and analysis, decision to publish, or preparation of the manuscript.

**Competing interests:** The authors have declared that no competing interests exist.

Development Goal of universal healthcare coverage that most countries are committed to [3]. One condition that summarises this threat is hypertension or high blood pressure (HBP), a condition which affects 22% of the population aged 18 years or older [1] and about 60% of individuals aged 60 years or older worldwide [4]. Yet, it is preventable and treatable, making it a specifically useful measure of healthcare system effectiveness [5].

Besides universal coverage, another critical objective of healthcare systems is to improve health equity, thus reducing inequalities in health according to socioeconomic status (SES), and in particular to wealth [6]. There exist significant health disparities according to SES in most MICs, with an epidemiological transition occurring during which the disease burden of NCDs is shifting from individuals with higher SES to individuals with lower SES [7]. Inequalities in access are often associated with the source of health insurance funding, partly driven by the large informal sector of the economy in MICs.

Although the successful prevention and treatment of NCDs among individuals from all socio-economic backgrounds present a litmus test for health systems in MICs [8], evidence on inequalities in NCDs in MICs and the potential effectiveness of healthcare systems in reducing the latter has at least two crucial shortcomings. First, few studies use objectively measured prevalence of NCD in general and HBP in particular, mostly relying on self-reported prevalence. This is problematic for two reasons. On the one hand, as a result of often poor health literacy among individuals with low SES, self-reported prevalence often overestimate their health status and are thus not aware of existing health conditions [9]. On the other hand, self-reports of HBP prevalence mostly relies on questions asking about conditions diagnosed by a doctor. This may lead to significant under-estimations of HBP prevalence among individuals from lower SES groups due to well-documented barriers in access to healthcare services among these groups. Second, although some studies have assessed the magnitude of inequalities in objectively measured HBP in MICs [10–21], little evidence exists on the factors associated with those inequalities, and especially the question whether healthcare systems effectively reduce inequities in health.

In this study, we use information from a group of seven MICs (China, Colombia, Ghana, India, Mexico, Russia and South Africa), corresponding to the years between 2007–2010 (2015 for Colombia), to assess: 1) levels of SES-related inequalities in (undetected and uncontrolled) HBP, 2) the association between HBP with individuals' socio-demographic characteristics as well as health insurance coverage, and 3) the contribution of different socio-demographic characteristics in explaining existing SES inequalities in HBP.

While undetected and uncontrolled HBP represent two inter-related outcomes of great importance for assessing wealth-related inequalities, given their direct relevance for HBP prevalence in the population, several mechanisms may explain SES-related inequalities therein. On the one hand, differences in (access to) material as well as immaterial resources, including income, transportation, time and health literacy between wealth groups, likely leads to individuals with lower SES having lower chances of being successfully diagnosed with HBP as well as having controlled HBP compared to individuals with higher wealth. For example, individuals with lower SES may lack the income to pay for transportation to see a doctor, while—at the same time—may also face significantly longer transportation times to the nearest doctor, compared to individuals with higher wealth. On the other hand, features of the healthcare system may exacerbate such SES-related inequalities in various ways. For example, lack of health insurance and co-payments for doctor visits and medicines often represent a significant barrier for effective healthcare coverage among individuals with lower wealth. Compared with individuals with higher SES, individuals with lower wealth are therefore less likely to consult a doctor, i.e., be diagnosed with HBP and less likely to be successfully treated for HBP. For the

aforementioned reasons, we hypothesise that—in all countries under study—lower SES is significantly associated with a lower likelihood of detected and controlled HBP.

## 2 Materials and methods

### 2.1 Data

The Study on Global Ageing and Adult Health (SAGE) project compiles studies on the population aged 60 and older. Countries included are China, Ghana, India, Mexico, Russian Federation and South Africa. We are using SAGE wave 0, which for China includes information for years 2007/08 and 2009/10; Ghana 2007/08; India 2007/08; México 2009/10; Russia 2007/08 and 2009/10; and South Africa 2007/08 [22]. Unfortunately, at the time of writing this study, we were not able to get access to the second wave studies collected at 2014/15. These surveys measure the determinants of active ageing through SES data, the physical and social environment, behaviour, cognition and affect, functionality, mental well-being, medical and health conditions, as well as use and access to health services.

For Colombia, the Healthcare, Welfare and Ageing Survey (SABE, for its acronym in Spanish) was collected in 2015, and its design is comparable to the other ageing population surveys [23]. Appendix A in S1 Appendix presents the main characteristics of the health systems of these countries.

One of the main characteristics of those surveys among older individuals is that they involve clinical measures such as blood pressure. Following a standardised procedure, nurses measured three times systolic and diastolic BP. After discarding the first BP measurement, we take as our objective measure of HBP whether the average of the measurements is above 140 mmHg for systolic BP (*SysBP*), or above 90 mmHg for diastolic BP (*DiasBP*). This is a standard procedure as the first BP measurement is usually higher than average because respondents tend to be nervous, a phenomenon known as the *white-coat effect*. In SABE, *SysBP* was measured in both arms (six measurements), while in SAGE it only was in one arm (three measurements). In Mexico, there were only two measurements, so we take the second one. Figure A1 in the S1 Appendix presents the densities of the final *SysBP* measure for each country. In this set of six countries, we observe an ample range of BP patterns: from the low levels of India to the high ones of South Africa.

We include in our analysis all observations of individuals aged 60 and older for whom there is available information in the following characteristics: SAGE studies include information for individuals 50 to 59 as well. We exclude them as SABE does not cover this age group. Nevertheless, online Appendix D in S1 Appendix presents the exercise using this sample, which are qualitatively the same as those presented in the main text. valid measurements of blood pressure, self-reported diagnosis of high blood pressure, gender, age, education, weight and height, smoking behaviour, assets, and health insurance (in all countries but Mexico). As a common limitation for using biomarkers data, the resulting sample is likely to be more educated, wealthier and care more for their health than the regular population [24, 25]. See Appendix B in S1 Appendix for further details, and the limitations under the discussion section for the implications of the selection pattern. Table 1 presents the means of the variables included, and sample sizes. In the rest of the document we explain the table and how variables are used. Individual sample weights are used in all exercises. These weights account for sample selection.

### 2.2 High blood pressure classification

Following Falaschetti et al. [26], we classify respondents in four groups, based on whether they are aware of having HBP and objective BP measures. Someone is considered **aware** if he or she reports having been diagnosed with HBP by a doctor or nurse, excluding women diagnosed

**Table 1. Means by country.**

| Variable | China | | Colombia | | Ghana | | India | | Mexico | | Russia | | South Africa | |
|---|---|---|---|---|---|---|---|---|---|---|---|---|---|---|
| | BPS | AR | BPS | AR | BPS | AR | BPS | AR | BPS | AR | BPS | AR | BPS | AR |
| **A. Proportions and means** | | | | | | | | | | | | | | |
| Population at risk | 0.67 | | 0.69 | | 0.57 | | 0.39 | | 0.77 | | 0.81 | | 0.81 | |
| HBP Aware | 0.33 | 0.49 | 0.57 | 0.82 | 0.15 | 0.26 | 0.18 | 0.46 | 0.43 | 0.55 | 0.61 | 0.75 | 0.37 | 0.46 |
| Undetected HBP | | 0.51 | | 0.18 | | 0.74 | | 0.54 | | 0.45 | | 0.25 | | 0.54 |
| Uncontrolled HBP | | 0.38 | | 0.35 | | 0.20 | | 0.20 | | 0.38 | | 0.61 | | 0.36 |
| Diastolic BP (mmHg) | 83.31 | 87.30 | 74.53 | 76.11 | 89.88 | 99.76 | 80.76 | 90.94 | 79.79 | 82.47 | 89.80 | 92.72 | 95.37 | 99.29 |
| Systolic BP (mmHg) | 145.52 | 156.28 | 133.79 | 140.00 | 138.65 | 153.89 | 124.90 | 140.79 | 146.92 | 154.08 | 146.24 | 151.47 | 148.44 | 154.59 |
| Male | 0.48 | 0.47 | 0.43 | 0.41 | 0.52 | 0.50 | 0.50 | 0.46 | 0.47 | 0.44 | 0.33 | 0.31 | 0.42 | 0.41 |
| Age | 68.94 | 69.41 | 70.29 | 71.20 | 71.02 | 70.80 | 68.19 | 68.63 | 69.76 | 70.06 | 71.60 | 72.08 | 69.00 | 68.91 |
| Obese (BMI $\geq$ 30) | 0.06 | 0.07 | 0.33 | 0.36 | 0.08 | 0.11 | 0.03 | 0.04 | 0.27 | 0.29 | 0.30 | 0.33 | 0.47 | 0.49 |
| Smoke ever | 0.35 | 0.34 | 0.51 | 0.49 | 0.26 | 0.23 | 0.57 | 0.54 | 0.44 | 0.42 | 0.22 | 0.19 | 0.33 | 0.32 |
| Education: Below Primary | 0.49 | 0.50 | 0.58 | 0.59 | 0.71 | 0.69 | 0.66 | 0.63 | 0.52 | 0.53 | 0.03 | 0.03 | 0.51 | 0.52 |
| Education: Primary | 0.22 | 0.22 | 0.30 | 0.30 | 0.09 | 0.09 | 0.16 | 0.15 | 0.27 | 0.27 | 0.10 | 0.10 | 0.22 | 0.22 |
| Education: Above Primary | 0.29 | 0.28 | 0.12 | 0.11 | 0.20 | 0.22 | 0.18 | 0.22 | 0.21 | 0.20 | 0.87 | 0.86 | 0.26 | 0.25 |
| Lives in urban area | 0.51 | 0.50 | 0.78 | 0.79 | 0.39 | 0.46 | 0.29 | 0.32 | 0.78 | 0.80 | 0.75 | 0.76 | 0.64 | 0.65 |
| Wealth index | 0.61 | 0.61 | 0.67 | 0.67 | 0.33 | 0.36 | 0.31 | 0.36 | 0.87 | 0.87 | 0.78 | 0.78 | 0.60 | 0.61 |
| Voluntary Health Insurance | 0.12 | 0.11 | 0.06 | 0.07 | 0.41 | 0.41 | 0.02 | 0.03 | | | 0.01 | 0.00 | 0.13 | 0.13 |
| No health insurance | 0.11 | 0.11 | 0.02 | 0.02 | 0.57 | 0.56 | 0.97 | 0.95 | | | 0.00 | 0.00 | 0.80 | 0.80 |
| **B. Number of observations** | | | | | | | | | | | | | | |
| All | 6689 | 4473 | 5228 | 3587 | 2496 | 1434 | 3454 | 1429 | 1238 | 893 | 2271 | 1827 | 1958 | 1572 |
| HBP Aware | | 2264 | | 2938 | | 358 | | 659 | | 513 | | 1496 | | 677 |
| Undetected HBP | | 2209 | | 649 | | 1076 | | 770 | | 380 | | 331 | | 895 |
| Uncontrolled HBP | | 1763 | | 1247 | | 272 | | 307 | | 367 | | 1182 | | 523 |

*Notes*: own calculations using SABE and SAGE studies with sample weights. Samples in the columns are defined as follows. **BPS**: those respondents for whom there is information on self-reported HBP and objective measurements. **At risk (AR)**: respondents who are either diagnosed with HBP (aware) or not diagnosed but with a systolic BP above 140 mmHg or a diastolic BP above 90 mmHg. The wealth index is defined per country, therefore it cannot be used to compare wealth between countries.

during pregnancy. The groups are defined as follows, and a diagram of these definitions presented in Fig 1. (i) **No HBP**: those respondents who declare not having being diagnosed with HBP and do not present objectively measured BP levels that indicate otherwise (*SysBP* < 140 and *DiasBP* < 90). (ii) **Undetected HBP**: those respondents who report not having HBP, but according to their BP measurements (*SysBP* $\geq$ 140 or *DiasBP* $\geq$ 90), they likely have the condition. (iii) **Uncontrolled HBP**: The person is *aware* of being diagnosed with HBP, but the BP measurements suggest that current treatment (if any) is ineffective. And (iv) **Controlled HBP**: those respondents who are *aware* of HBP, and whose BP levels are under control.

In this study, the denominator of our analysis are individuals who either report having ever been with HBP, or those whose BP is above the standard diagnosis threshold (groups ii to iv). Henceforth, we refer to them as the **at-risk population (AR)**. While group (i) is essential for public health interventions to prevent them from developing HBP, in this work, we concentrate on those who already require attention by the health system. In this sub-population, it is possible to discuss the false negatives of the system (group ii). False-positives, those diagnosed with HBP but who did not have the condition, cannot be detected with this survey-based approach. This group is important as it indicates the inefficient use of resources of the system. The other leading group of interest is (iii), as it involves those for whom the health system has

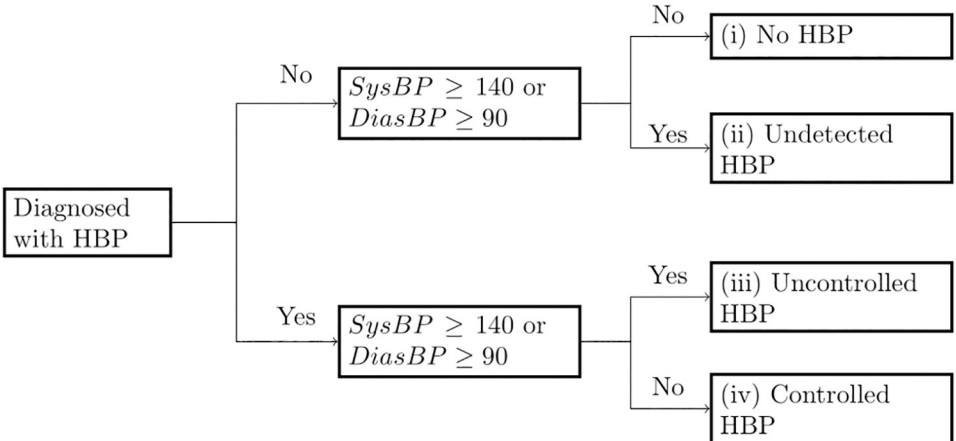

**Fig 1. Definition of high blood pressure classification groups.** *Notes*: Diagnosis of HBP by a physician correspond to the self-response of individuals. *SysBP* and *DiasBP* correspond to the blood pressure measurement by a nurse at the time of the survey.

not provided effective treatment. Ideally, these two groups should be zero. Another element of interest is the proportion of those who are aware (groups iii and iv) and are receiving treatment. We concentrate on an outcomes perspective, rather than on an input perspective. Therefore we make emphasis in the controlled/uncontrolled partition. Thus, our main dependent variables will be:

- to have undetected HBP for those individuals in the population AR;

- to have uncontrolled HBP conditional on being aware of the HBP condition.

### 2.3 Socio-economic status / wealth index

In order to understand the role of SES on our HBP variables, we constructed a **wealth index** based on assets' ownership. This is a composite measure of a household's cumulative living standard. This index measures wealth based on household's ownership of selected assets and is built using factor analysis. Details of the weights given to each variable are presented in the online Appendix B in S1 Appendix. The index was constructed with the following variables: the household appliances that counted the home, type of dwelling, housing rights (ex: own house, rent), physical characteristics of the dwelling (roofing, walls and floors material). In the Colombian case, we also included a socio-economic level indicator known as *estrato* (according to a utility subsidies and taxes classification). Fig 2 presents estimated densities of the index for each country. The value of the index is relative for each country: our objective is to understand the concentration of wealth, rather than it's level. As shown in the graph, India and Ghana are in one extreme, while Russia is in the other. All the other countries are relatively similar in that respect.

### 2.4 Discrete choice models

In order to approximate the relationship between and individual being classified as with undetected HBP with household wealth, and between belonging to the group uncontrolled HBP with household wealth, we estimate logistic regression models given that in both cases the dependent variable is binary. The probability of observing that for individual $i$ outcome $Y_i$ take

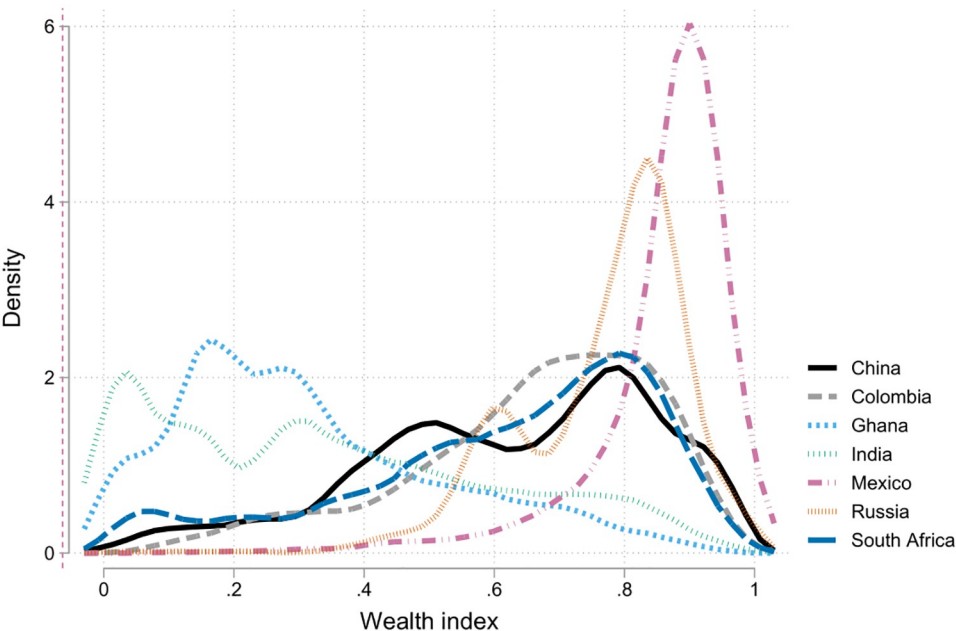

**Fig 2. Densities of the wealth index by country.** *Notes*: Epanechnikov Kernel densities using a bandwidth of 0.03 (asset index ranges from 0 to 1). Graph constructed over the sample of SABE and SAGE of respondents with a valid BP measure and a self-report of HBP. The asset index was derived using factor analysis for all individuals who answered the assets section. See online Appendix B.2 in S1 Appendix for further details.

the value of 1 instead of 0, conditional on a set of covariates $X_i$, country $R_i$, and wealth index $wealth_i$ is given by the following equation:

$$Pr(Y_i = 1|X_i) = \Lambda \left( \sum_{r=1}^{6} (\alpha_r + \gamma_r \cdot \text{wealth}_i + X_i \beta_r) \cdot 1[R_i = r] \right) \qquad (1)$$

where $\Lambda$ represents the logistic function. For the case of uncontrolled HBP, the sample is restricted to those diagnosed with the condition. In this regression, on top of the countries fixed effects and a dummy for respondents being surveyed in 2009/10, we include the interaction between the country dummies and the independent variables. As a result, the interaction of the wealth index and the country dummy provides a measure of the gradient ($\gamma_r$) for each country. The comparison group is being surveyed on 2007/08. The Colombian survey was the only one in 2015, so its information is already part of the specific survey fixed effect. These models were estimated using Stata 16.

From the results of these models, we compute the average marginal effects (AME) of wealth on the respective probabilities: $E\left[\frac{\partial Pr(Y_i=1|X_i,R_i)}{\partial wealth_i}\right]$. This expectation can be computed conditional on the country. The resulting figures can be interpreted as increases on percentage points (pp) over the probability of $Y = 1$ as opposite to $Y = 0$. The odds ratio, an alternative for interpreting the coefficients of the model, are presented in the Table C1 in S1 Appendix.

A generalisation of the model is the multinomial logistic model. This model fits the context as our at-risk group is composed of three categories. The model considers the probability of occurrence of two of the three events relative to a basis. We present in the Table C1 in S1 Appendix the relative risk ratios (rrr), similar to the odds ratio in the binary case.

## 2.5 Blinder Oaxaca decomposition

The Blinder Oaxaca decomposition technique is a common tool for explaining inequality of health outcomes [27]. We use it to study mean outcome differences in uncontrolled and undetected High blood pressure (HBP) by poverty status. We considered as poor to 20% of people with the lower income. Sensitivity to alternative cutoffs is presented in the online Appendix G in S1 Appendix. This method divides the uncontrolled or undetected HBP differential between two groups (poor and non-poor) into a part that could be explained by group observable characteristic differences and a residual part that cannot be accounted for by such differences in HBP determinants. The unexplained part subsumes the effects of group differences in unobserved predictors.

We used a twofold decomposition using the coefficients from a pooled linear probability model over both groups as the reference coefficients. Then, the outcome difference is divided into two components:

$$R = Q + U \tag{2}$$

where the first component,

$$Q = \{E(X_{poor}) - E(X_{non-poor})\}'\beta^* \tag{3}$$

is the part of the outcome differential that is explained by group differences in the predictors. As predictors, we used behavioural risks which include the variables of age and being male, demographic risks, which have smoking history and obesity status, and a variable that indicates if the individual lives in an urban area. The second component in this decomposition,

$$U = E(X_{poor})'(\beta_{poor} - \beta^*) + E(X_{non-poor})'(\beta^* - \beta_{non-poor}) \tag{4}$$

is the unexplained part, which in addition to capture all the potential effects of differences in unobserved variables can be a proxy of inequality in HBP by poverty status.

Fairlie [28] consider an extension of the original decomposition but using non-linear probability models such as the logit and probit. Under this framework, the contribution of each variable to the gap is equivalent to the change in the average predicted probability if the distribution of the poor is replaced with the distribution of the non-poor. According to the authors, the linear probability model may produce notoriously different covariates that have strongly non-linear relationships with the outcome. Therefore, we compute the decomposition under this extension in the Appendix C in S1 Appendix. These models were estimated using the command Oaxaca in Stata 16 [29].

## 3 Results

### 3.1 High blood pressure groups and wealth

Panel A of Table 1 presents the means of the relevant variables conditional on different samples (for which there is no missing data in all covariates). Columns marked with BPS (Blood Pressure Sample), corresponds to those individuals for whom there is information on their blood pressure. Given that, we can determine the proportion of total respondents who are AR, which is presented in the first row. In Russia and South Africa, at the time of the SAGE interviews between 2007–10, around 81% of respondents aged 60 or older were AR, followed by Mexico (77%) and Colombia (69%). Below was China (67%), and far from them Ghana (57%) and India (39%). The second row presents those who were aware of having HBP. If the denominator is the entire population (columns BPS), we have the detected prevalence, which was as high as 61% in Russia and as low as 15% in Ghana around 2007–10.

The other set of columns of Table 1, present the proportions for the population AR. For instance, China had a detected prevalence of 33% on the overall 60 and older population. These individuals were nearly half (49%) of those at risk, the AR population. Thus, the undetected BP rate was 51%. Relative to the AR population, 12% had their BP levels controlled (*SysBP/DiasBP* BP below 140/90 mmHg). Thus, the proportion of individuals aware, but with their BP being uncontrolled, corresponded to 39% (= 51%—12%) of the AR population at the time (around 2007–10). This is the uncontrolled HBP rate. Panel B of Table 1 presents the total number of observations for each of these groups.

Concerning undetected HBP there were four groups. First, in Colombia and Russia, undetected HBP rates were 18% and 25%, respectively. Second, for Mexico the corresponding rate was 45%. Third, for China, India, and South Africa it was around 53% at the time. And fourth, in Ghana, this figure was just above 74%. Once we consider control of the condition, the best performing country was India: 20% of Indians AR had their *SysBP* below 140 mmHg and their *DiasBP* below 90 mmHg at the time. The worst performing country was Russia, where 61% of the AR were known cases of HBP which are not under control at the time.

Table 1 presents as well socio-economic characteristics of the respondents, corresponding to the years 2007–10 for all countries except Colombia (for which data correspond to 2015). Age and gender produce 'mechanical' differences between countries. However, has fewer men in the sample (33%), yet BP levels were one of the highest at the time, in comparison with the other countries included in the sample; the South African population was on average as old as the Indian (68 years), but there were more than ten mmHg of difference between their average BP levels. Differences were more likely to be related to habits and lifestyle: obesity in South Africa was nearly 50%, while in India it was 4% at the time. Yet, China obesity rate was as well low, 7%, but their average *SysBP* was the same in South Africa. Mexico and Colombia had the highest rate of people living in an urban area, close to 80%. However, BP in Mexico was higher than in Colombia. Although Mexicohad the highest wealth index also had one of the highest BP in our sample of countries. Another consideration is that there were no substantial systematic differences between the BPS and AR population because AR represented a large proportion of all individuals older than 60 in these countries.

Another set of variables presented in Table 1 is related to the health system. As described in the context section, there are noteworthy differences. Russia had no VHI at all at the time, while in Ghana, 41% of the individuals had it. Almost the rest of the population is uninsured in Ghana, only worse to India, where nine out ten did not have such protection at the time. The opposite is again Russia, where everyone was insured, similar to Colombia, where people without health insurance represented only just 2% in 2015. A very similar pattern occurs for living in rural areas, where more barriers have been found to expand universal coverage in the world. Unfortunately, this information is not available for Mexico.

Socio-economic characteristics, lifestyle, and health system differences shape the relationship between wealth and health in each country. The unconditional relationship between wealth and our HBP quality of attention indicators are presented in Fig 3. Each point of the line presents the average rate of undetected and uncontrolled HBP for each level of wealth around the years 2007–2010 (and 2015 for Colombia). First, it shows a negative slope between wealth and probability of non-detection of HBP, which depends on the country. Second, concerning uncontrolled HBP, there are both positive and negative slopes, showing that the role of wealth is context-dependent. With the AME derived from the parametric model presented above, we can determine whether the observed slopes hold after conditioning of different sets of controls.

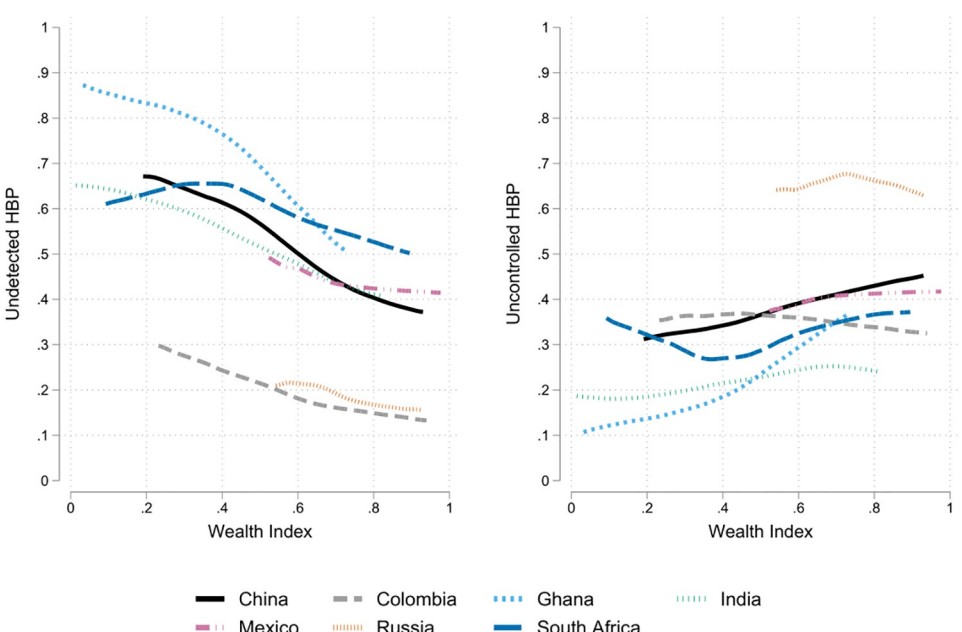

**Fig 3. Undetected and uncontrolled HBP and wealth.** *Notes*: Kernel-weighted local polynomials using the samples of SABE and SAGE. For each country/region, the domain corresponds to the range covering 5%-95% of the total density of the wealth variable.

## 3.2 Results from discrete choice models

Table 2 presents the AME after logistic regressions. It considers three models for each outcome which differ on the set of controls. Columns (1) and (5) only consider country fixed effects, age, and a gender dummy. Columns (2) and (6) expand them to lifestyle: whether the individual is obese and if she has ever smoked. These are the main risk factors, and there are not so many differences between both sets of columns. Columns (3) and (7) include education and whether the respondent resides in an urban area. Finally, columns (4) and (8) consider health insurance schemes. These are characteristics that might be correlated with wealth. Thus the resulting coefficients correspond to the remaining disparities not associated with these variables.

If we consider only variables not related to wealth, we find that concerning undetected HBP, there has a wealth gradient in all countries (column 2). Ghana and Russia are the most extreme cases: if wealth has ten levels, moving up in one level is associated with a reduction of 7.27 pp the probability that someone above the age of 60 had undetected hypertension in Ghana, and of 7.64 pp in Russia, conditional on risk factors such as obesity and smoking antecedents. If we consider education level on top of these controls (column 3), the gradient drops to 6.64 pp in Ghana and 7.45pp in Russia. South Africa is on the opposite scenario, where the gradient was around 2.05 pp without education and 1.66 pp (not significant at 90% level) when these controls are added. Health system variables reduce the gradient slightly in most countries, especially in China. The considerable reduction on the gradient for Russia has to be taken carefully, as less than 1% of the individuals at risk has VHI, and none had no insurance. Therefore the change is such point estimate obeys more to the high imprecision of the estimate (a 95% confidence interval between -6.28 pp and -83.11 pp).

Regarding uncontrolled HBP, Table 2 presents a similar analysis but conditional on the diagnosis. Here we are exploring the probability of having HBP in comparison with a mix

**Table 2. Average marginal effects of wealth on undetected and uncontrolled HBP by country.**

| Country | $\frac{\partial \text{Undetected}}{\partial \text{Wealth}}$ | | | | $\frac{\partial \text{Uncontrolled}}{\partial \text{Wealth}} \vert \text{Diagnosis} = 1$ | | | |
|---|---|---|---|---|---|---|---|---|
| | (1) | (2) | (3) | (4) | (5) | (6) | (7) | (8) |
| China | -0.585 *** | -0.558 *** | -0.491 *** | -0.297 *** | -0.343 *** | -0.342 *** | -0.322 *** | 0.024 |
| | (0.034) | (0.035) | (0.041) | (0.055) | (0.056) | (0.058) | (0.064) | (0.083) |
| Colombia | -0.255 *** | -0.255 *** | -0.265 *** | -0.258 *** | -0.257 *** | -0.268 *** | -0.266 *** | -0.166 ** |
| | (0.036) | (0.037) | (0.043) | (0.055) | (0.051) | (0.052) | (0.055) | (0.071) |
| Ghana | -0.748 *** | -0.727 *** | -0.664 *** | -0.580 *** | -0.303 ** | -0.288 ** | -0.272 ** | -0.244 |
| | (0.059) | (0.063) | (0.074) | (0.099) | (0.135) | (0.131) | (0.126) | (0.180) |
| India | -0.458 *** | -0.453 *** | -0.308 *** | -0.250 *** | -0.239 ** | -0.271 ** | -0.296 ** | -0.261 * |
| | (0.062) | (0.065) | (0.072) | (0.090) | (0.107) | (0.117) | (0.124) | (0.142) |
| Mexico | -0.629 *** | -0.648 *** | -0.694 *** | | -0.346 | -0.426 | -0.192 | |
| | (0.167) | (0.177) | (0.148) | | (0.347) | (0.346) | (0.436) | |
| Russia | -0.821 *** | -0.764 *** | -0.745 *** | -0.447 ** | -0.348 ** | -0.408 *** | -0.382 *** | -0.441 ** |
| | (0.190) | (0.190) | (0.231) | (0.196) | (0.150) | (0.152) | (0.148) | (0.184) |
| South Africa | -0.219 ** | -0.205 ** | -0.166 | -0.049 | -0.112 | -0.115 | -0.082 | -0.005 |
| | (0.096) | (0.095) | (0.105) | (0.115) | (0.120) | (0.123) | (0.123) | (0.114) |
| Observations | 15127 | 15127 | 15127 | 14223 | 8767 | 8767 | 8767 | 8247 |
| Age and gender | X | X | X | X | X | X | X | X |
| Obesity and smoking | | X | X | X | | X | X | X |
| Education, urban | | | X | X | | | X | X |
| Health Insurance | | | | X | | | | X |

*Notes*: own calculations using SABE and SAGE studies with individual sample weights. Average marginal effects after logistic regressions are presented in the table. In each estimated equation, each country was multiplied by the wealth index in order to obtain a specific gradient. Controls differ according to columns: (i) age and being male; (ii) obesity status and smoking history; and (iii) education level (primary, and above primary), living in an urban area, having voluntary health insurance, and not having health insurance at all. All regressions include country dummies, a dummy that indicates that the individual was surveyed in year 2009/10 as opposite to 2007/08, and the interaction between each control and this set of dummies. Household level clustered standard errors are presented in parentheses. Significance:

* 0.1,

** 0.05,

*** 0.01

between those people with controlled HBP and those who are not aware of the condition. In the Appendix F in S1 Appendix, we present results without conditioning on the diagnosis. The table shows a significant negative gradient for all countries but for Mexico and South Africa. For these countries, the gradient (around the years 2007–10) was negative as well, but the precision of the estimates cannot rule out the non-presence of an association. The largest figure is for Russia (-0.41 in column 6) and Mexico (-0.42, but not significant at 95% level), and the smallest for South Africa (-0.11). Notice that the dispersion between countries is smaller than for detection. In most cases, the addition of health insurance variables reduces the magnitude of the gradient, especially for China where the gradient disappears.

Estimated odd ratios of the models (3) and (6) are presented in Table C1 in the S1 Appendix, using the same controls as columns 3 and 7 of Table 2, as well as regressions for *SysBP* and an alternative multinomial logistic model. Regardless of the model specification, and even with the continuous *SysBP* variable, similar patterns to the ones presented above are obtained. This is also the case when alternative definitions of the age range (Appendix D in S1 Appendix) and the assets index (Appendix E in S1 Appendix); and when individual regressions per country and set of controls are considered (Appendix H in S1 Appendix).

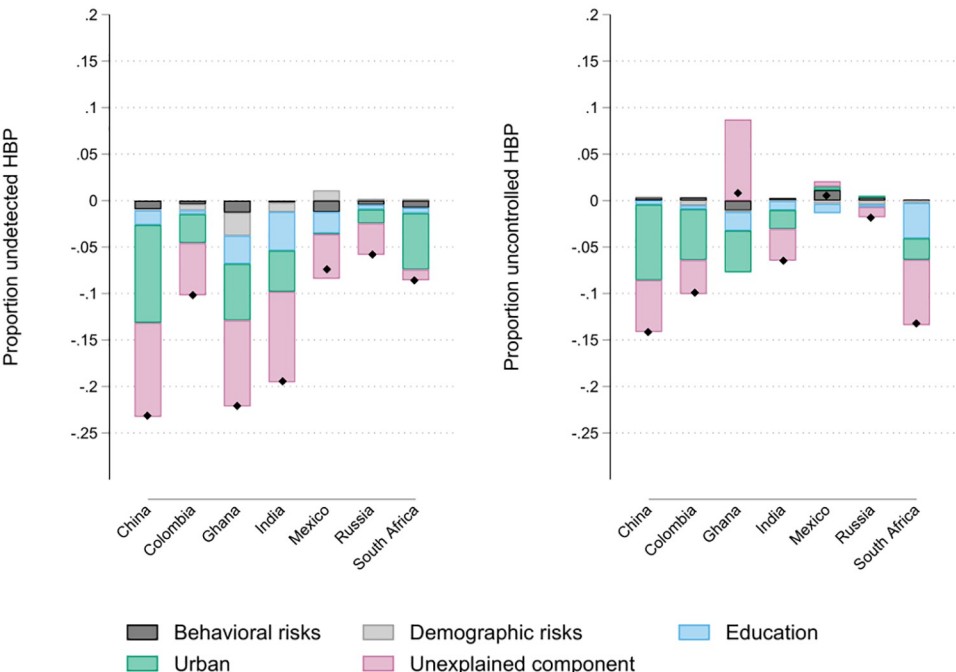

**Fig 4. Contribution to gap in undetected and uncontrolled HBP conditional on being aware o hypertension.**
**Notes**: Contribution of covariate groups to the gap on undetected and uncotrolled HBP between individuals in the lowest wealth quintile agains the other quintiles. These numbers are computed with the Blinder-Oaxaca decomposition after linear probability models.

### 3.3 Decomposition results

We further explore the gradient by comparing differences between those at the bottom of the wealth distribution (the lowest quintile) with the rest of the sample. The black diamonds in Fig 4 present the difference in the proportion of undetected HBP (on the left) and uncontrolled HBP (on the right). Concerning the previous section, the pattern for the overall gap differs in the case of Russia. For this country, there exists a small gap in this exercise while a large wealth gradient. This indicates that the gradient at the time was probably linked to a specific upper interval of the domain of the wealth index (where most of the distribution is located as shown in Fig 2), and not a feature that is constant across the wealth distribution. For Ghana, the large gap goes in line with the steep gradient.

For this exercise, we focus on decomposition analysis. For example, for China the gap between *poor* and *rich* on undetected HBP was of -23.3 pp. This number is obtained from taking the undetected HBP rate of 44.9% for the 80% of the sample with higher wealth and subtracting the 68.1% rate for the lower quintile (see Table C2 in the S1 Appendix). For this alternative view of the gradient, we can decompose the contribution of each of the observed factors -behaviours, demographics, education and urban/rural division- using the Blinder-Oaxaca method. For the case of China, 56.2%(= -0.131/-0.233) of the gap at the time can be explained with the difference in the level of the observed variables. This is roughly the same proportion for most countries (from 45% to 58.4%), apart from Mexico and South Africa. In Mexico, it was 34.4% of a gap of 7.32 pp which is not significant at the 95% level, and in South Africa, it is 86.32% of the gap of 8.41 pp can be explained with the observed variables.

Almost all the explained differences are due to the urban/rural division in the first place and the education level in the second. Again, for the case of China, differences in the

proportion of people living in urban areas represented 80.15%(= -0.105/-0.131) of the explained difference at the time; this figure is also high for South Africa (83.5%), and very small in Mexico (3.4%). Differences in education level represent another 11.8% in China, and just an 8% for the rest of the variables (socio-demographics, smoking prevalence and obesity). In Mexico, 91.3% of all the explained gap can be associated with education. In general, these two factors accounted for most of the observed gap (more than 80% of it). However, in Ghana, age and gender differences represented nearly 11.3% of the explained difference. In Figure C1 in the S1 Appendix, we perform the decomposition separately by urban and rural areas. It shows that all the other observed characteristics played a minor role in explaining wealth gaps within these areas 2007–10. For Ghana, the number of observations with a positive diagnosis of HBP in the rural setting is too small for the analysis.

Tables C5 in the S1 Appendix includes the health insurance variables into the decomposition. The proportion of the gap explained is similar to the results above. These variables explain 14% of the difference in Ghana, 11.6% in South Africa, 10.6% in India, 9.4% in Colombia, and 2.5% in Russia at the time. In contrast, for China (-3.5%) the level of observable variables suggest that the gap should run in the opposite direction.

The absolute difference in uncontrolled HBP, conditional on the diagnosis, between poor and non-poor people is statistically significant for China, Colombia and South Africa (see Table C3 in the S1 Appendix). For the other countries, the total difference cannot be rejected to be equal to zero. For Ghana, India, Mexico, and Russia, there is no evidence of a gap though there was evidence of a gradient. Once again, this might be related to the point of comparison of the wealth distribution. Ghana presents interesting results, as the point estimate is small, but the explained component suggests a gap similar to the Colombian or the Indian. For this country, any analysis should be careful as the number of individuals diagnosed with HBP is small, hence results are very imprecise.

For the three countries for which there is evidence of a gap, the explained component is roughly between 50% and 60%. For China and Colombia, more than 90% was explained by rurality, while this factor is only 36% in South Africa. In the last country, education accounts for 77% of the explained gap, while it is 6% in China and Colombia. When health insurance variables are included (Table C5 in the S1 Appendix), there are almost no changes at all, and these variables explain around 4% to 6% of the explained gap.

As discussed in the methods section, the BO decomposition with linear probability models might be biased for covariates with a strong non-linear relationship with the outcome. In the S1 Appendix, Tables C6 and C7 show the results of the Fairlie extension based on a logit model [28]. Results are almost the same as those presented above. Moreover, as for the results of the marginal effects, we consider alternative definitions of the age range (Appendix D in S1 Appendix), the assets index (Appendix E in S1 Appendix), and the uncontrolled HBP results conditional on being aware of the conditions (Appendix E in S1 Appendix). We also consider the sensitivity of the results to alternative definitions of the non-poor sample (Appendix G in S1 Appendix). Qualitative results are stable across these alternatives.

## 4 Discussion

In this study, using data for 2007–10 and 2015 for Colombia, we explored the role of household wealth on the quality of seven MICs' health systems for attending mature patients aged 60 and older with HBP, both aware and not of their condition at that time. This condition, which requires an effective action from several layers on the primary health care system, is central for preventing the onset of cardiovascular diseases.

As presented by Maimaris et al. [30], treating HBP has four inter-related goals: awareness, treatment, antihypertensive medication adherence and control. Despite differences in health systems, there existed several common elements for attaining these goals: access to health insurance, low prices/co-payments for medication, consistent access to health care. Over these areas, the MICs analysed in this study, at the time the data were collected, presented important differences:

- The population at risk was between 67% and 81% in China, Colombia, Mexico, Russia and South Africa, compared to the range of 39%—57% in Ghana and India. This is potentially linked with the higher urbanisation rates of the first countries at the time. Obesity also might have played an essential role for Colombia, Mexico and Russia.

- Russia and Colombia, in 2007–2015, already had reasonable detection procedures, which allowed them to have high awareness rates (75%-82%). Ghana, around the same time, did a poor job with 26%, and in other countries included in this study, only half of those at risk were aware of their condition.

An important indicator of health system effectiveness is the degree to which it reduces wealth-related inequalities in health. As the results of this study show, in all of the MICs included, there were significant wealth-related differences in the likelihood of having detected HBP, with individuals with lower wealth being less likely to be diagnosed than those with higher wealth. To explore the roots of these gradients, we considered the following factors: demographic (age and gender), behavioural (obesity and smoking), socio-economic (education and living in urban areas), and health system (access to compulsory and voluntary health insurance). We found that:

- Wealth was essential in all countries for determining awareness, but it was especially relevant in Ghana and Russia. In all countries, the wealth gradient was less pronounced once we control for access to HI.

- Once diagnosed with HBP, the wealthier the individuals, the more likely their BP was under control. For China, the gradient disappeared once health insurance is under control.

- The most relevant observed factor for the gap on undetected and uncontrolled rates was to live in a rural area rather than an urban area. It is substantially more important for most countries than education level, the second determinant in the list. Differences in demographic characteristics and behavioural risks played a small role in explaining large differences in most countries. This is likely because the wealth gap on these variables was relatively small, rather than that these variables were not associated with the outcomes.

Health insurance played a different role according to the context. In Ghana, South Africa and India, it accounted for around 10% of the explained gap. In such countries, there were large proportions of the population without any health insurance. In contrast, in Russia and China, these variables explained little of the gap and have small uninsured populations and the VHI sector. Colombia is the exception, as it also had a small VHI sector and few people uninsured, but these variables can explain up to 9.4% of the gap at the time (2015). Russia and Colombia provide different versions of universal coverage: the two have high levels of awareness and very different wealth gradients. Colombia, in 2015, had a gradient that cannot be explained with any of the observed factors, and Russia had a sharp gradient that drops dramatically when access to VHI is considered (yet, it is imprecisely estimated). The reasons for such divergence need further research to be fully understood. Another difference is the period: Russia surveys are from 2007 to 2010, while the Colombian one from 2015. However, our

comparison is between the state of two health systems based on control of HBP. There were no many differences in terms of the technology available or their prices between these two periods.

For two countries, we can present a comparison with the literature. First, for China, the gradient in prevalence was non-existent, and that there was a positive gradient between detection and control [18]. Second, for South Africa, results are in line with Thomas et al. [11] who found no socio-economic gradient in unawareness once controls are considered. They use a large sample that involves respondents of all ages and estimate the gradients using a finite-mixture model that accommodates unobserved heterogeneity.

For the case of keeping under control the BP of patients who are aware of their condition, wealth gradients were negative as well. Interestingly, its magnitude is the same even for countries such as Colombia and Russia, which had large awareness of the condition at the time. A potential reason is a composition effect: countries first have to detect HBP, but it becomes increasingly complicated to treat the 'marginal' patient. This composition effect is one of the explanations given by Mackenbach [31] for persistent health inequalities in Nordic countries. Despite their generous welfare states, poorer individuals in rich societies usually engage in behaviours linked to bad health due to a persistent difference in immaterial resources. For instance, preferences for unhealthy habits, physical activities, nutrition, etc. The behavioural component does not play a prominent role in our estimates, which probably highlights a limitation of the available measurements for capturing differences in lifestyle. For the rest of the countries, where large proportions of the population aged 60 and older are under risk, gradients are either non-existent or not pronounced enough to be detected. Keeping under control BP of detected individuals is not a trivial problem as it goes beyond the system's coverage. Individual behaviour plays a central role. Patients might improve -or not- their health-related behaviour to early detection of the condition and its treatment [32–34].

It is important to mention that the results presented are associations; they cannot be understood as causal effects between wealth and health. In other words, it cannot be concluded that increasing the wealth of the *current* population will result in smaller probabilities of detection of HBP. As seen in the decomposition, at the time of the data was collected (2007–2010 and 2015 for Colombia, there was a large share of the gap explained by other contextual characteristics related to both variables (in particular, place of residence and education). Also, poor health (which might be correlated with HBP onset) might result in poverty. Exercises that assess the impact of wealth on health require specific exogenous shocks. A good example is Frijters et al. [35], who using the reunification of Germany, found only a small impact of income on health satisfaction. Furthermore, although the information on blood pressure is objectively measured and based on nationally representative surveys, information on doctor diagnosis with HBP is self-reported. Due to recall issues the latter could result in an underreporting of diagnosis with HBP, especially among individuals with lower SES and possibly lower levels of cognitive functioning. If so, in tendency, this would downward bias our estimates of inequalities in HBP detection treatment. Unfortunately, however, as a result of lacking administrative data on doctor diagnoses with HBP, we are unable to assess this empirically, and we are unaware of previous studies from MICs able to do so either. Finally, it is important to keep in mind that the data used in this study only represent a snapshot that reflects the situation in 2007–10 (2015 for Colombia). Especially given the expansion of primary healthcare in most MICs, socio-economic inequalities in HBP detection and treatment may have decreased since then. However, if this indeed has been the case needs to be investigated.

## Supporting information

**S1 Appendix.**
(PDF)

## Author Contributions

**Conceptualization:** Philipp Hessel, Paul Rodríguez-Lesmes.

**Data curation:** María Fernanda García, Paul Rodríguez-Lesmes.

**Formal analysis:** María Fernanda García.

**Funding acquisition:** Paul Rodríguez-Lesmes.

**Methodology:** Philipp Hessel, Paul Rodríguez-Lesmes.

**Project administration:** Paul Rodríguez-Lesmes.

**Software:** María Fernanda García, Paul Rodríguez-Lesmes.

**Validation:** Philipp Hessel.

**Visualization:** María Fernanda García, Paul Rodríguez-Lesmes.

**Writing – original draft:** Philipp Hessel, Paul Rodríguez-Lesmes.

**Writing – review & editing:** Philipp Hessel, Paul Rodríguez-Lesmes.

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
