## [Decision Letter · Decision Letter 0]

21 Jan 2022

PONE-D-21-31685Wealth and inequality gradients for the detection and control of hypertension in older individuals in middle-income economiesPLOS ONE

Dear Dr. Rodríguez-Lesmes,

Thank you for submitting your manuscript to PLOS ONE. After careful consideration, we feel that it has merit but does not fully meet PLOS ONE’s publication criteria as it currently stands. Therefore, we invite you to submit a revised version of the manuscript that addresses the points raised during the review process.

We look forward to receiving your revised manuscript.

Kind regards,

Rajiv Janardhanan, Ph.D.

Academic Editor

PLOS ONE

Journal Requirements:

GADC project by the CIHR/IDRC [grant

number 108442-001]

Fulbright-Colciencias and Colombia Cientifica – Alianza EFI 60185 contract FP44842- 220-2018, funded by The World Bank through the Scientific Ecosystems, managed by the Colombian Ministry of Science, Technology and Innovation (MINCIENCIAS)

This project was funded under the GADC project by the CIHR/IDRC [grant number 431

108442-001], and the program Fulbright-Colciencias and Colombia Cientifica – Alianza 432

EFI 60185 contract FP44842- 220-2018, funded by The World Bank through the 433

Scientific Ecosystems, managed by the Colombian Ministry of Science, Technology and 434

Innovation (MINCIENCIAS).

GADC project by the CIHR/IDRC [grant

number 108442-001]

Fulbright-Colciencias and Colombia Cientifica – Alianza EFI 60185 contract FP44842- 220-2018, funded by The World Bank through the Scientific Ecosystems, managed by the Colombian Ministry of Science, Technology and Innovation (MINCIENCIAS)

Reviewers' comments:

Reviewer's Responses to Questions

**Comments to the Author**

1. Is the manuscript technically sound, and do the data support the conclusions?

Reviewer #1: Partly

Reviewer #2: Yes

2. Has the statistical analysis been performed appropriately and rigorously? 

Reviewer #1: I Don't Know

Reviewer #2: No

3. Have the authors made all data underlying the findings in their manuscript fully available?

Reviewer #1: Yes

Reviewer #2: Yes

4. Is the manuscript presented in an intelligible fashion and written in standard English?

Reviewer #1: Yes

Reviewer #2: Yes

5. Review Comments to the Author

Reviewer #1: The SAGE data that had been used to study nearly 13 to 14 years old. it is better present a research articles using recent data rather than old data. Nearly 14 years have gone many changes would have occurred in MIC countries. concluding results using old data is not fair.

Reviewer #2: (i) The data collection derived from different countries are taken in the basis of the survey using SES data. How can these data can give rise to statistical study? Just surveys can’t be accepted as data.

(ii) The information about the second group in the study population “Undetected HBP” is not clear in the methodology. How do you differentiate group 1 (No HBP) and group 2 (Undetected HBP)? Because the definition of both the above groups were similar (Page no. 3).

(iii) Based on the above comment 2, In the statistical table (Table2), the comparative study between the undetected and uncontrolled HBP is questionable (???).

(iv) In the estimation of relationship between undetected and uncontrolled HBP, the use of the logistic regression model should be justified (Page no. 4).

(v) The Oaxaca (or Blinder-Oaxaca) decomposition usually used to show the differences in an outcome variable across groups or over time can be separated into explained and unexplained portions.

In what way, the above method reflects in this study to find outcome differences in uncontrolled and undetected HBP? It should be justified.

6. PLOS authors have the option to publish the peer review history of their article (what does this mean?). If published, this will include your full peer review and any attached files.

Reviewer #1: No

Reviewer #2: No

---

## [Author Response · Author response to Decision Letter 0]

26 Mar 2022

Date: March 3rd, 2022

Re: Manuscript revision

Dear Dr. Rajiv Janardhanan

We very much thank you for the opportunity to revise our manuscript, originally titled ”Wealth and inequality gradients for the detection and control of hypertension in older individuals in middleincome economies”. The detailed comments and suggestions by the two reviewers have been extremely helpful and incorporating them has greatly improved the accessibility of the paper. Below, we respond in detail to each comment.

As a result of the reviewing process, we have done a major revision to the writing of the document so the key methodological decisions are clearer for a multidisciplinary audience. We very much hope that the substantial changes made to the manuscript are in line with the feedback from the reviewers and remain fully open to any further suggestions.

Sincerely,

The Authors 

1 Review Comments to the Author (in the email)

1. Reviewer #1: The SAGE data that had been used to study nearly 13 to 14 years old. It is better present a research articles using recent data rather than old data. Nearly 14 years have gone many changes would have occurred in MIC countries. Concluding results using old data is not fair.

Response: We thank the reviewer for flagging-up the question regarding the antiquity of the surveys used for the analyses. It is absolutely true that the data from SAGE are already around 13-14 years old and therefore may very well not represent the current situation anymore.

However, a central aim of this study was to perform a comparative study of several middleincome countries in order to highlight differences between them. As our analyses reveal, at the time the SAGE and SABE data were collected, important differences between countries existed with regard to treatment and control of high blood pressure, and socio-economic inequalities therein. The existence of these important differences -we would argue- does provide an argument in favor of a comparative design as showing and quantifying these differences also highlights the potential for interventions, e.g. by the healthcare system, to improve diagnosis and treatment of high blood ressure.

Against this background, to our knowledge, unfortunately no more up-to-date comparable data source exists that contains information on objectively measured high blood pressure, which is an essential dimension for our study. Hence, there exists a clear trade-off between timeliness and the ability to compare several countries.

Although our analyses only represent the situation around 2007-2010 (and 2015 for Colombia) they provide an important point of comparison for any future study being able to use new (comparable) data on the subject in order to assess whether health inequalities in high blood pressure have decreased or increased since then.

While fully agreeing with the reviewer that the previous version did not sufficiently make clear that the data used are from around 13 years ago in all parts of the manuscript, we have now re-written the title, the abstract and main body of the manuscript in order to always make clear that the analyses refer solely to the moment of time that the data were collected (see red-line copy of the revised manuscript).

Changes to manuscript:

• As mentioned above, we now mention in the abstract, title and discussion that the data are from 2007-10 (2015 for Colombia).

• Furthermore, we have changed the tense from present when referring to the empirical results, which could imply that the results refer to the current moment, to past tense in order to make clear that the results refer to a point in time in the past.

• We also added the following two sentences at the very end of the manuscript (in the discussion section) stating that: ”Finally, it is important to keep in mind that the data used in this study only represent a snapshot that reflects the situation in 2007-10 (2015 for Colombia). Especially given the expansion of primary healthcare in most MICs, socioeconomic inequalities in HBP detection and treatment may have decreased since then. However, if this indeed has been the case needs to be investigated.”

2. Reviewer #2: (i) The data collection derived from different countries are taken in the basis of the survey using SES data. How can these data can give rise to statistical study? Just surveys can’t be accepted as data.

Response: We very much thank the Reviewer for this comment, leading us to consider the use of survey data for analyzing socio-economic inequalities in health. We fully agree that survey data are by far perfect and that superior data exist, at least in principle, e.g. in the form of register data. However, such data often do not exist in many Lower- and Middle-Income Countries. While for some countries information on morbidity, in the form of hospital discharges, exists, administrative information - that is representative at the country-level - for chronic conditions, such as high blood pressure diagnosis, does not exist. Furthermore, similar to previous studies on the subject, is interested in quantifying socio-economic inequalities not only in detected (i.e. doctor diagnosed) instances of high blood pressure but also in inequalities in undiagnosed high blood pressure. Information on the latter is not available through administrative data, not even by health insurance providers, since many individuals with high blood pressure have never been diagnosed by a doctor (for various reasons, including lack of health insurance coverage).

We would like to stress that the literature on socio-economic inequalities in health, from disciplines including demography, economics, epidemiology and public health, routinely uses survey data for its purposes:

• Hajizadeh, M., Karen Campbell, M., Sarma, S. (2014). Socioeconomic inequalities in adult obesity risk in Canada: trends and decomposition analyses. The European Journal of Health Economics, 15(2), 203-221.

• Njagi, P., Arsenijevic, J., Groot, W. (2020). Decomposition of changes in socioeconomic inequalities in catastrophic health expenditure in Kenya. PloS one, 15(12), e0244428.

Furthermore, many peer-reviewed and published studies on the topic use the same SAGE data, e.g.:

• Kunna, R., San Sebastian, M., Stewart Williams, J. (2017). Measurement and decomposition of socioeconomic inequality in single and multimorbidity in older adults in China and Ghana: results from the WHO study on global AGEing and adult health (SAGE). International journal for equity in health, 16(1), 1-17.

• Kailembo, A., Preet, R., Williams, J. S. (2018). Socioeconomic inequality in self-reported unmet need for oral health services in adults aged 50 years and over in China, Ghana, and India. International journal for equity in health, 17(1), 1-14.

• Brinda, E. M., Kowal, P., Attermann, J., Enemark, U. (2015). Health service use, outof-pocket payments and catastrophic health expenditure among older people in India: The WHO Study on global AGEing and adult health (SAGE). J Epidemiol Community

Health, 69(5), 489-494.

• Sadana, R., Blas, E., Budhwani, S., Koller, T., Paraje, G. (2016). Healthy ageing: raising awareness of inequalities, determinants, and what could be done to improve health equity. The Gerontologist, 56, S178-S193.

• Brinda, E. M., Attermann, J., Gerdtham, U. G., Enemark, U. (2016). Socio-economic inequalities in health and health service use among older adults in India: results from the WHO Study on Global AGEing and adult health survey. Public Health, 141, 32-41.

Including also studies published in PLoS One, e.g.:

• Harttgen, K., Kowal, P., Strulik, H., Chatterji, S., Vollmer, S. (2013). Patterns of frailty in older adults: comparing results from higher and lower income countries using the Survey of Health, Ageing and Retirement in Europe (SHARE) and the Study on Global AGEing and Adult Health (SAGE). PLoS One, 8(10), e75847.

Although self-reports of chronic conditions, including high blood pressure, can be problematic, e.g. due to poor health literacy, an advantage of our study is precisely that we use objectively measured blood pressure to categorize individuals into whether or not they actually have high blood pressure.

• Finally, we would like to stress that all SAGE surveys, as well as the one for Colombia (SABE), are based on stratified random samples and representative on the country-level. (See: Kowal, Paul, et al. ”Data resource profile: the World Health Organization Study on global AGEing and adult health (SAGE).” International journal of epidemiology 41.6 (2012): 1639-1649.)

Changes to manuscript: We now mention explicitly in limitations that this information is selfreported. Hence, we state that: ”Furthermore, although the information on blood pressure is objectively measured and based on nationally representative surveys, information on doctor diagnosis with HBP is self-reported. Due to recall issues the latter could result in an underreporting of diagnosis with HBP, especially among individuals with lower SES and possibly lower levels of cognitive functioning. If so, in tendency, this would downward bias our estimates of inequalities in HBP detection treatment. Unfortunately, however, as a result of lacking administrative data on doctor diagnoses with HBP, we are unable to assess this empirically, and we are unaware of previous studies from MICs able to do so either.”

3. Reviewer #2: (ii) The information about the second group in the study population “Undetected HBP” is not clear in the methodology. How do you differentiate group 1 (No HBP) and group 2 (Undetected HBP)? Because the definition of both the above groups were similar (Page no. 3).

Response: These two groups are in fact mutually exclusive, and the data allows differentiating both groups as the survey has information on both systolic and diastolic blood pressure, on top of the diagnosis of high-blood pressure (HBP). Hence, ”No HBP” refers to individuals that were not diagnosed with HBP by a doctor and at the same time do not have objectively measured systolic or diastolic blood pressure levels above the respective thresholds. In contrast, ”Undetected HBP” refers to individuals that do have objectively measured systolic or diastolic blood pressure levels above the respective thresholds but have not been diagnosed with HBP by a doctor.

Changes to manuscript: We have added a diagram (new Figure 1) and amended the explanation of the definitions to make them clearer.

4. Reviewer #2: (iii) Based on the above comment 2, In the statistical table (Table2), the comparative study between the undetected and uncontrolled HBP is questionable (???).

Response: As these are mutually exclusive groups, the direct comparison is possible. The comparison ends up being for individuals who report not having been diagnosed with HBP, between those with blood pressure levels below and above the standard thresholds.

Changes to manuscript: None.

5. Reviewer #2: (iv) In the estimation of relationship between undetected and uncontrolled HBP, the use of the logistic regression model should be justified (Page no. 4).

Response: We are using a logistic regression model (logit) as the dependent variables are binary, as described at the end of section 2.2. In general applied econometrics work there are little differences when considering average marginal effects of either logit, probit, and linear probability models.

Changes to manuscript: We have amended the first paragraph of section 2.4 so it is clear that we are considering binary dependent variables.

6. Reviewer #2: (v) The Oaxaca (or Blinder-Oaxaca) decomposition usually used to show the differences in an outcome variable across groups or over time can be separated into explained and unexplained portions. In what way, the above method reflects in this study to find outcome differences in uncontrolled and undetected HBP? It should be justified.

Response: In our analysis the dependent / outcome variables are (i) belonging to the uncontrolled HBP group or not, and (ii) belonging to the undetected HBP group or not. The ’groups’ that the reviewer mentions of the BO decomposition correspond to the wealth of households (poor and non-poor). This type of use is common in studies aiming to measure the extend of inequality is several type of outcomes, diving it between a proportion of the difference between the means that can be attributed to observed characteristics (other than poverty status), and non-observed characteristics. Some examples of their use in the field of health inequalities are: • Rahimi, E., & Hashemi Nazari, S. S. (2021). A detailed explanation and graphical representation of the Blinder-Oaxaca decomposition method with its application in health inequalities. Emerging Themes in Epidemiology, 18(1), 1-15.

• Liao, P. A., Chang, H. H., Wang, J. H., & Sun, L. C. (2016). What are the determinants of rural-urban digital inequality among schoolchildren in Taiwan? Insights from BlinderOaxaca decomposition. Computers Education, 95, 123-133.

• Sharaf, M. F., & Rashad, A. S. (2016). Regional inequalities in child malnutrition in Egypt, Jordan, and Yemen: a Blinder-Oaxaca decomposition analysis. Health economics review, 6(1), 1-11.

• Emamian, M. H., Zeraati, H., Majdzadeh, R., Shariati, M., Hashemi, H., Fotouhi, A. (2014). Economic inequality in eye care utilization and its determinants: a Blinder–Oaxaca decomposition. International Journal of Health Policy and Management, 3(6), 307.

Changes to manuscript: We have edited the first paragraph of section 2.5 in order to add the requested justification.

2 Comments in PDF document attached to email

1. Page 1: Title: wealth and inequality gradients are associated factors and not the factors for detection and control of hypertension. The title has to (be) modified. Model that was used by the author can be included in the title for prediction of wealth and inequality in detection and control of hypertension.

Response:

• New title perhaps simply ”Socio-economic inequalities in the detection and control of high-blood pressure in six low- and middle-income countries around 2007-2015”

Changes to manuscript:

2. Page 3: The SAGE wave data used for the study is 13to 14 years old. many changes have developed in the health system after 2010. in these countries. for example in India national level program was implemented for non communicable diseases and geriatric population of the rural populuation / poor socioeconomic status free of cost and health insurance of 0.5 million per year for below poverty line population. thus assuming all the counties of MIC as similar is not acceptable for using prection model.

Response: As discussed in our response to Reviewer #1, it is absolutely true that the data does not represent the current situation anymore if the aim were to provide an status of the control of HBP across countries. Yet, our objective is different: our objective was to highlight differences between countries (in a given period of time) with regard to treatment and control of high blood pressure across socio-economic groups. For this, we needed data that included objective information on blood pressure control, and unfortunately there is no updated information publicly available for them. Such differences illustrate what elements are important to consider for low and middle-income countries in general, and are no specific policy recommendations to the current state of each country. In this new version we hope that for the reader is clear that we are working with data from a specific time-framework.

As for the comment on ”assuming all the countries of MIC as similar is not acceptable for using prediction model” [’prediction’, we understood], we completely agree with this. We are not assuming countries are similar. We show differences in context by country and also run models separately by country. We do not use the model for predicting the future of HBP detection and control in each of the countries, but once again, to provide general ideas of the set of features that a country might consider for the expansion of their health systems.

Changes to manuscript: The title, abstract, introduction, and discussion in general, were modified to reflect the limitations.

3. Page 4: what is the advantage of this model compared to estimation of yield in population which is more simpler model every decison and policy makers can understand.

Response: Unfortunately we are not quite sure of what the reviewer means by ’estimation of yield in population’. What we can say is that (i) the logistic regression is the most standard methodology for analysing the ’determinants’ of binary outcomes (in both economics and health sciences), and (ii) the Blinder-Oaxaca model is the most standard decomposition tool that assess the role of each determinant on the difference of an outcome between groups. This last tool is slightly less common, but it is an standard in public health and epidemiology (Rahimi & Nazari, 2021).

The advantage of using regression models, on top of reviewing unconditional means (which we also do by analysing Table 1), is that conditional means can isolate competing stories by ’controlling’ for differences in other variables that can potentially explain the incidence of poverty and of uncontrolled/undetected HBP.

Changes to manuscript: We have amended both the first paragraphs of sections 2.2 and 2.5 in order to justify better the model choice.

4. Page 9: This statement contradict your hypothesis that uncontrolled hypertension was significantly associated with lower economic status. But at risk population is higher in urban population.

Response: Unfortunately the reference to page 9 alone is a bit vague, and we do not know exactly which statement the reviewer is referring to. Section 3.2 indeed shows that for some countries that hypothesis might not hold (Mexico and South Africa).

Changes to manuscript: None

3 References

Rahimi, E., & Hashemi Nazari, S. S. (2021). A detailed explanation and graphical representation of the Blinder-Oaxaca decomposition method with its application in health inequalities. Emerging Themes in Epidemiology, 18(1), 1-15.

---

## [Decision Letter · Decision Letter 1]

16 May 2022

Wealth and inequality gradients for the detection and control of hypertension in older individuals in middle-income economies around 2007-2010

PONE-D-21-31685R1

Dear Dr.

We’re pleased to inform you that your manuscript has been judged scientifically suitable for publication and will be formally accepted for publication once it meets all outstanding technical requirements.

Kind regards,

Rajiv Janardhanan, Ph.D.

Academic Editor

PLOS ONE

Additional Editor Comments (optional):

Reviewers' comments:

Reviewer's Responses to Questions

**Comments to the Author**

1. If the authors have adequately addressed your comments raised in a previous round of review and you feel that this manuscript is now acceptable for publication, you may indicate that here to bypass the “Comments to the Author” section, enter your conflict of interest statement in the “Confidential to Editor” section, and submit your "Accept" recommendation.

Reviewer #2: All comments have been addressed

Reviewer #3: All comments have been addressed

2. Is the manuscript technically sound, and do the data support the conclusions?

Reviewer #2: Yes

Reviewer #3: Yes

3. Has the statistical analysis been performed appropriately and rigorously? 

Reviewer #2: Yes

Reviewer #3: Yes

---

## [Editor Report · Acceptance letter]

27 May 2022

PONE-D-21-31685R1 

Wealth and inequality gradients for the detection and control of hypertension in older individuals in middle-income economies around 2007-2015 

Dear Dr. Rodríguez-Lesmes:

I'm pleased to inform you that your manuscript has been deemed suitable for publication in PLOS ONE. Congratulations! Your manuscript is now with our production department. 

Kind regards, 

on behalf of

Dr. Rajiv Janardhanan 

Academic Editor

PLOS ONE